# IFT20: An Eclectic Regulator of Cellular Processes beyond Intraflagellar Transport

**DOI:** 10.3390/ijms232012147

**Published:** 2022-10-12

**Authors:** Francesca Finetti, Anna Onnis, Cosima T. Baldari

**Affiliations:** Department of Life Sciences, University of Siena, 53100 Siena, Italy

**Keywords:** intraflagellar transport system, IFT20, primary cilium, vesicular traffic

## Abstract

Initially discovered as the smallest component of the intraflagellar transport (IFT) system, the IFT20 protein has been found to be implicated in several unconventional mechanisms beyond its essential role in the assembly and maintenance of the primary cilium. IFT20 is now considered a key player not only in ciliogenesis but also in vesicular trafficking of membrane receptors and signaling proteins. Moreover, its ability to associate with a wide array of interacting partners in a cell-type specific manner has expanded the function of IFT20 to the regulation of intracellular degradative and secretory pathways. In this review, we will present an overview of the multifaceted role of IFT20 in both ciliated and non-ciliated cells.

## 1. Introduction

Almost 30 years ago, the discovery of the intraflagellar transport (IFT) system by Rosenbaum’s lab paved the way to the characterization of the molecular pathways that regulate the assembly and maintenance of cilia and flagella [1]. Cilia and flagella are hair-like structures with locomotive properties (motile cilia and flagella) or the ability to sense the surrounding environment and regulate signaling pathways (non-motile cilia or primary cilia). In humans, the primary cilium (PC) is found in almost all cell types, whereas motile cilia are restricted to a group of specialized cells as exemplified by cells of the respiratory tract [2]. Flagella are found primarily on gametes.

At the structural level, these cellular organelles are composed of a central part consisting of an axoneme of nine microtubules (nine plus two central microtubules for motile cilia and flagella, nine plus zero lacking the central pair for the non-motile PC) that extends from a basal body (a modified mother centriole) surrounded by the ciliary membrane, a specialized extension of the plasma membrane [3,4,5]. Cilia and flagella are devoid of ribosomes, and all proteins required in these organelles must be imported from the cell body. This process relies on an active transport system, known as the intraflagellar transport (IFT) system [1,6,7,8].

Initially discovered in 1993 in the unicellular green alga *Chlamydomonas reinhardtii*, further studies demonstrated that the IFT is operational in most eukaryotic cilia and flagella (reviewed in [9,10]). Along the length of longitudinal sections of Chlamydomonas flagella, the IFT system appears as granule-like particles localized between the microtubules and the flagellar membrane, forming a long series of particles referred to as IFT trains [1,11]. The IFT machinery consists of two biochemically distinct IFT complexes, IFT-A and IFT-B complex, containing different proteins. Specifically, the IFT-A complex contains six proteins (IFT43, IFT121, IFT122, IFT139, IFT140 and IFT144), and the IFT-B complex 16 proteins (IFT20, IFT22, IFT25, IFT27, IFT38, IFT46, IFT52, IFT54, IFT56, IFT57, IFT70, IFT74, IFT80, IFT81, IFT88 and IFT172) [12,13] (Figure 1). The main role of the IFT complexes is to carry cargos along the axonemal microtubules from the cell body to the tips of cilia and flagella (anterograde transport) and then back to the cell body (retrograde transport) with the help of molecular motors kinesin and dynein, respectively, for cilia assembly, disassembly and maintenance [14,15,16,17,18]. In recent years, new functions for IFT have been linked to signaling pathways that regulate key cellular functions, including, among others, Hedgehog, Wnt and Notch, and to the traffic of ciliary membrane components, including membrane receptors such as G-protein-coupled receptors (GPCRs), receptor tyrosine kinases (RTKs), the Sonic hedgehog (SHH) receptor Patched1 (PTCH1), and a variety of ion channels and signaling proteins [19,20,21,22,23,24,25,26,27,28]. In addition to its roles in ciliated cells, the IFT system participates in vesicular trafficking also in non-ciliated cells [29].

Among the IFT proteins, the smallest is IFT20, the only IFT protein with a dual localization (basal body and Golgi) in ciliated cells that confers this protein pleiotropic functions. Here, we review the cumulative knowledge of IFT20, starting from its structure and function to discuss the most recent findings on its role in the assembly of the non-motile primary cilium and vesicular trafficking. Moreover, other emerging roles of IFT20 in both ciliated and non-ciliated cells will be discussed.

## 2. Molecular Dissection of IFT20: Structure and Localization

In humans, the *IFT20* gene is composed of five exons and is localized on chromosome 17q11.2. Four isoforms have been described for IFT20, produced by alternative splicing, and five additional potential isoforms have been computationally mapped. Among the described isoforms, isoforms 2, 3 and 4 differ from the canonical one (isoform 1) for the insertion of 26 or 16 amino acids for isoform 2 and 3, respectively, or deletion of a stretch of 18 amino acids for isoform 4. The canonical isoform of human IFT20 is highly conserved among mammals and the other classes of vertebrates (fish and amphibians), with an 85–99% amino acid sequence homology as well as a functional homology as a regulator of ciliary processes. Among invertebrates, IFT20 is only found in *D. melanogaster* and *C. elegans,* with a 33% and 36% homology with human IFT20, respectively. Notwithstanding the lower homology, IFT20 is required for ciliogenesis also in nematodes [30].

Until now, the only domain identified in the human IFT20 structure is the coiled-coil (CC) domain at position 74–114 at its C-terminus, which mediates protein–protein interactions with other CC domain-containing components of the IFT complex [31,32] and potentially with other CC domain-containing proteins. Recently, a motif in the sequence of IFT20, corresponding to amino acids Y111-E118-F124-I129 (YEFI motif) has been identified as responsible for binding to the WD40 domain of the autophagy-related protein ATG16 [33].

IFT20 is expressed in wide variety of tissues with the highest expression in testis, liver, pancreas and adrenal gland [34]. At the subcellular level, IFT20 localizes at the primary cilium and at the basal body of ciliated cells [9]. The basal body localization of IFT20 is cholesterol-dependent, as shown by the failure of IFT20 to be recruited to the centrioles when intracellular cholesterol distribution was perturbed by depletion of the peroxisomal protein TMEM135 [35]. During the cell cycle, IFT20 associates with centrosomes [36]. Moreover, IFT20 is the only IFT protein that was also found associated with both the Golgi apparatus and early and recycling endosomes [36,37] in non-dividing cells, revealing new functions for IFT20 other than intraflagellar transport.

## 3. Ciliary Trafficking of IFT20: From the Golgi Apparatus to the PC Tip

### 3.1. Cilium Assembly and Maintenance

The PC is a cellular structure with unique protein and lipid distribution whose composition is finely regulated in time and space [38,39,40]. There is no evidence that ciliary proteins may be directly synthesized in the PC. Instead, proteins localized in the PC are transported to the ciliary base from the Golgi and the cytosol by vesicular trafficking. Trafficking is then taken over by the IFT system.

IFT20 is anchored to both the cis and medial cisternae of the Golgi complex by the Golgi resident protein GMAP210, a member of the golgins [41] (Figure 2). A pool of the CC domain-containing 41 (CCDC41) associates with IFT20 at the Golgi and participates in the formation and transport of ciliary vesicles from the Golgi complex to the basal body [42]. These vesicles are enriched in ciliary membrane proteins to be transported to the PC [43]. The release of IFT20 from the Golgi is mediated by another Golgi-resident protein, GM130, which forms a complex with VPS15 and allows the release of IFT20-associated vesicles with the contribution of the microtubule-based motor KIFC1 to ensure the IFT20-dependent transport of membrane proteins toward the PC [44,45,46].

IFT20-associated vesicles interact and fuse with vesicles decorated by the small GTPase Rab8 and its effector Rabaptin5, and are directed to the base of the PC with the help of IFT54 (elipsa in zebrafish) that functions as a bridge between IFT20 and Rabaptin5 [32]. The presence of Rab8 in these vesicles ensures the transport of vesicles to the PC and their fusion with the plasma membrane at the ciliary base [38,47]. Rab8 acts as a membrane trafficking regulator in the Rab11-Rab8 cascade in which Rabin8, the guanine nucleotide exchange factor for Rab8, binds to Rab11 and is delivered to the centrosome. At the centrosome, Rabin8 associates with the multimolecular complex Bardet–Biedl syndrome complex (BBSome) on vesicles to activate Rab8 to promote ciliary membrane assembly [39,40]. The proteins EHD1 and EHD3 interface with this cascade to allow the recruitment of transition-zone proteins and IFT20 during centriole maturation and its transformation into the basal body [48]. EHD1/EHD3 depletion by RNA interference results in a defective accumulation of IFT20-positive vesicles at the mother centriole with defective ciliogenesis. A centrosomal pool of CCDC41 interacts with IFT20 and promotes its recruitment to the centrosome to mediate docking of ciliary vesicles to the mother centriole [42].

### 3.2. Trafficking of Receptors and Signaling Molecules

The mosaic of IFT20 interactors contributes not only to guide vesicle transport from the Golgi apparatus and to the basal body but also provides specificity to the trafficking of IFT20 cargos. Moreover, the discovery that other IFT subunits, as for example IFT88, are associated with vesicles different from those marked by IFT20 implies that different IFT proteins play complementary roles in delivering the unique types of cargos associated with the respective vesicles [35,49]. An elegant study by Monis VJ et al. helps to elucidate how ciliary receptors take different routes. IFT20, in concert with GMAP210 and the exocyst complex, directs the vesicular transport of transmembrane proteins such as fibrocystin and polycystin-2, while delivery of the G protein-coupled receptor Smoothened is largely independent of these components [50]. Moreover, IFT20 is responsible for the basal body localization of pallidin, one subunit of the biogenesis of lysosome-related organelles complex-1 (BLOC-1) responsible for the ciliary delivery of polycystin-2, but not Smoothened or fibrocystin, through the recycling trafficking pathway [50]. IFT20 may also indirectly regulate the localization of the receptor PDGFRα through association with the E3 ubiquitin ligases c-Cbl and Cbl-b and is required for Cbl-mediated ubiquitination and internalization of PDGFRα for feedback inhibition of receptor signaling [51].

Recently, a new interactor of IFT20 in ciliated cells has been identified as the autophagy-related protein ATG16, which as part of this complex carries out an autophagy-unrelated function. Specifically, the interaction of ATG16 with IFT20 is required for the transport of the phosphatase INPP5E to the PC to ensure the correct local phosphoinositide distribution, on which the function of the PC crucially depends [33].

## 4. Cilia-Unrelated Vesicular Trafficking

Consistent with its high structural homology with membrane coat components revealed by bioinformatic analyses [52] and its localization in cytoplasmic vesicular compartments in non-ciliated cells, such as T cells [53], secondary retinal neurons [54], keratinocytes [55] or breast cancer cells [56], IFT20 has been involved in extraciliary processes that rely on vesicular trafficking. Indeed, IFT20 is required for the orchestration of signaling by surface receptors, cellular degradation pathways, secretion and glucose metabolism (Figure 3).

### 4.1. Trafficking of Receptors and Signaling Molecules

The role of IFT20 in intracellular trafficking has been well characterized in T lymphocytes. Consistent with its association with a number of vesicular compartments, including Rab5+ early endosomes and Rab4+ or Rab11+ recycling endosomes, IFT20 selectively controls intracellular transport in T cells acting in concert with other components of the IFT system. IFT20 is required to achieve specificity within the widely used recycling network orchestrated by Rab GTPases, which label recycling endosomes containing specific receptors, by coordinating a recycling pathway that is exploited by the T cell antigen receptor (TCR) and transferrin receptor (TfR), but not by the chemokine receptor CXCR4 [37,57,58,59]. The general mechanism of recycling posits that, following internalization into Rab5+ early endosomes, recycling receptors are delivered back to the cell membrane in recycling endosomes using a short loop or a long loop that depends on Rab4 or Rab11, respectively [60]. It was found that in the absence of IFT20, recycled TCRs accumulate in Rab5+ vesicles, with a concomitant reduction in Rab4+ and Rab11+ compartments, suggesting a role for IFT20 in the traffic of internalized receptors from early to recycling endosomes [37].

Constitutive TCR recycling acts as a quality control mechanism that allows the identification and degradation of defective TCRs, but also permits the formation of an intracellular pool of TCRs that plays a pivotal role during T cell activation by ensuring long-lasting signaling. The engagement of plasma membrane-associated TCRs by peptide-loaded major histocompatibility complex (pMHC) on the surface of an antigen-presenting cell (APC) triggers the establishment of a transient cell polarity with the assembly of a specialized structure known as immunological synapse (IS). IS assembly relies on an extensive reorganization of receptors, signaling mediators and components of cytoskeleton as well as on the recruitment of the intracellular pool of TCRs at the IS [61]. IFT20 has been found to accumulate at the IS, where it regulates polarized TCR recycling [53]. Furthermore, loss of IFT20 is associated with decreased TCR signaling and defective recruitment to the IS of the membrane-associated adaptor protein linker for activation of T cells (LAT), which serves as a scaffold for the assembly of multimolecular signaling complexes that amplify and diversify the signals triggered by the TCR [62]. Interestingly, the delivery of LAT+ vesicles to the IS is dependent on the golgin GMAP210, with which IFT20 associates at the Golgi, although the IFT20-binding activity of GMAP210 appears not to be required for the polarized transport of LAT [63].

Importantly, as in the IFT particles, IFT20 cooperates with other IFT proteins, including IFT88, IFT52, IFT54 and IFT57 as well as with newly identified IFT20-binding partners (i.e., ARPC3 and ERGIC-53), to control TCR recycling in T lymphocytes [37,53,64]. In addition, IFT20, IFT52 and IFT57 act together within the same pathway to control polarized recycling to the IS of the TfR, but not CXCR4 [37]. These data indicate that IFT20 selectively regulates intracellular trafficking in T cells, acting in a cooperative manner with IFT proteins and other specific interactors. Consistent with the key role of IFT20 in IS assembly, conditional *Ift20*-deficient mice show defective antigen-specific T-cell responses due to defective proinflammatory cytokine production in mouse models of T-cell-driven colitis and collagen-induced arthritis [62,65].

In agreement with its role in T lymphocyte polarization during IS assembly, IFT20 was found to be involved in cell reorganization during migration [55,56,66,67]. In non-ciliated breast cancer cells, IFT20 has been reported to interact at the trans-Golgi network with two negative regulators of migration, Numb and Ctnnal1, assisting their transport to the plasma membrane. Consistent with this function, loss of IFT20 in the non-ciliated mouse breast cancer cells enhances epithelial–mesenchymal transition, lamellipodia formation and cell migration. Additionally, IFT20 interacts with the actin-associated protein Tagln2, which limits cell migration and whose expression is downregulated in IFT20-deficient breast cancer cells, which may further contribute to their enhanced migration [56]. In contrast, IFT20 has been shown to promote invasiveness and migration of osteoblastoma and colorectal cancer cells independently of the PC by promoting the reorientation of the Golgi complex toward the direction of migration, the anterograde Golgi transport and the GM130–AKAP450 interaction to nucleate Golgi-associated microtubules, suggesting a cell-specific role for IFT20 in this process [66,67]. Of note, IFT20 also regulates the intra-Golgi transport of surface-exposed membrane type 1-matrix metalloproteinase (MT1-MMP) in human osteosarcoma cells [67]. Since MT1-MMP has been shown to play a pivotal role in the formation and function of the invadopodia, which are responsible for matrix degradation [68], this finding indicates that IFT20 might be implicated in the invasive program of carcinoma cells.

By contrast, Su et al. have recently demonstrated that IFT20 is not involved in Golgi polarization but participates in polarized β1-integrin recycling to focal adhesions during migration of epidermal cells [55]. More specifically, they found that IFT20 deficiency is associated with the entrapment of β1 integrin in Rab5+ endosomes during focal adhesion turnover in keratinocytes, similar to what has been found for the TCR in T cells [37].

Similarly to IFT20, IFT88 has been reported to play a cilia-independent role in the polarization of migrating kidney cells [69], and knockout of IFT and ciliary proteins leads to embryonic developmental alterations associated with defective cell motility [70]. Together, these findings indicate that IFT20 promotes the intracellular trafficking events required for cell polarization. This occurs both at the PC and at homologous structures that share structural and functional similarities, such as the IS and focal adhesions, where cell surface receptors, signaling molecules and cytoskeletal components are recruited in a polarized manner [55,71]. Accordingly, IFT20, together with IFT52 and IFT57, also participates in polarized vesicular trafficking toward the postsynaptic dendritic terminals in secondary retinal neurons [54].

The IFT complex B proteins IFT20 and IFT88 have also been reported to play a role independent of the PC in the modulation of the Hippo signaling pathway during cardiogenesis [72]. The Hippo pathway is involved in cardiac development and regeneration in vertebrates [73]. IFT20 and IFT88 associate with the Hippo signaling transcriptional coactivator YAP1/WWTR1 (TAZ), and IFT88 modulates YAP1 activity by sequestering it away from the nucleus to restrict proepicardium and myocardium development [72]. These data indicate a non-canonical role for IFT20 in heart development, in addition to its previously reported function in regulating Hedgehog signaling at the cardiac PC [74].

### 4.2. Cellular Degradation Pathways

New functions of IFT20 in intracellular membrane trafficking related to the cellular degradation pathways have been reported in ciliated cells as well as in the non-ciliated T and B lymphocytes [75]. One of these pathways is autophagy, which Pampliega et al. have shown to be regulated by IFT20 in ciliated cells [76]. Autophagy is a vesicle-based catabolic process that delivers proteins and organelles to lysosomes, in order to generate basic structural components and energy required to maintain cellular and tissue homeostasis [77,78]. IFT20 is a target of autophagy and acts in concert with IFT88 in a signaling pathway involving Hedgehog by promoting the transport of some components of the autophagic machinery to the ciliary base during cell starvation [76]. Autophagy is required for the selective degradation of IFT20 under nutrient-rich conditions, which prevents ciliary growth. This process is reverted in conditions of serum deprivation, when stress-induced autophagy is activated and IFT20 is spared from degradation and can promote ciliary growth and the autophagic process. In this pathway, IFT20 interacts with the phagophore elongation complex component ATG16L and is required for its vesicular trafficking to the basal body in ciliated cells [76,79]. More recently, the association of IFT20 with ATG16L1 has been also described in T lymphocytes [80]. In these cells, IFT20 promotes the association of ATG16L1 with the Golgi complex and early endosomes as well as autophagosome biogenesis from Rab5+ vesicles. IFT20 targets ATG16L1 to the early endosome-associated BECLIN 1 complex, allowing for the recruitment of downstream autophagy regulators to promote T cell autophagy.

In addition, recent data indicate that IFT20 is important for a late step in the autophagic pathway, namely lysosomal degradation, in both ciliated and non-ciliated cells [81]. IFT20-deficient T cells exhibit a lysosomal defect due to impaired intracellular traffic of lysosomal hydrolases. These enzymes are targeted to lysosomes by a vesicular pathway mediated by the cation-independent mannose-6-phosphate receptor (CI-MPR). IFT20 interacts with both the CI-MPR and the microtubule motor dynein, as well as with the retromer component VPS35, to promote the dynein-dependent retrograde transport of the CI-MPR to the trans-Golgi network. Consistent with the implication of IFT20 in CI-MPR retrograde trafficking, lysosomal targeting of hydrolases is compromised by IFT20 depletion [81]. Therefore, loss of IFT20 is relevant to lysosome functionality and has been associated with a compensatory enhancement of lysosome biogenesis, a process dependent on the CLEAR gene network coordinated by the transcription factor TFEB. Accordingly, TFEB activity is increased in IFT20-deficient T cells due to reduced mTOR signaling, which regulates the nuclear translocation or TFEB. The lysosome-related function of IFT20 is shared by other non-ciliated cells, such as B lymphocytes, and ciliated cells [81]. Consistently, it has been demonstrated that sperms from a male germ cell-specific *Ift20*−/− mouse show defective degradation of cytoplasmic components of the residual bodies due to the reduction in the number of mature lysosomes [82].

### 4.3. Secretion

In addition to its function in signaling and degradative pathways, IFT20 has been implicated in procollagen secretion by acting on ER-to-Golgi trafficking in cranial neural crests, multipotent stem cells essential for the formation of craniofacial skeletal components [83]. Interestingly, IFT20 was found to promote skull development both through its canonical role in PC assembly and function by mediating PDGF signaling required for osteogenic proliferation and cell survival, and through a non-canonical activity in the secretion of type I collagen. Loss of IFT20 in osteoblasts is associated with defective mineralization due to delayed secretion of type I collagen, suggesting that IFT20 may be implicated in ER export complexes [83]. Interestingly, a similar phenotype has been associated with mutations in the gene *sec23a*, which encodes a component of the coat protein complex II involved in anterograde protein trafficking from the ER to the Golgi apparatus and with which the IFT proteins have striking structural homologies [52]. Mutations in this gene cause cranio-lenticulo-sutural dysplasia, a disorder characterized by skeletal defects and late-closing fontanels together with hypertelorism [84,85]. In addition, GMAP210, with which IFT20 is associated at the Golgi complex, promotes both anterograde and retrograde trafficking between the ER and the Golgi apparatus, and its mutations cause neonatal lethal skeletal dysplasia [44,86,87]. In other craniofacial tissues, in particular during condylar cartilage development, IFT20 is required for Golgi apparatus expansion in chondrocytes during the secretion of cartilaginous matrix [88]. Importantly, IFT20 plays a critical role in controlling bone formation not only during embryogenesis, but also in adults, as witnessed by the osteopenia-like phenotypes reported in the skull in the absence of IFT20 [89]. Furthermore, during bone formation, IFT20 has been implicated in post-translational modifications of collagen lysines involved in collagen cross-linking by modulating the expression of the ER chaperone FKBP65 associated with the telopeptide lysyl hydroxylase 2, presumably via a ciliary dependent function of IFT20 [89]. Recently, it has been reported that IFT20 deletion in osteoblast lineage reduces bone strength and stiffness due to defective cell alignment and polarity during bone development, acting on ceramide-pPKCζ-β-catenin signaling [90].

### 4.4. Glucose Metabolism

Similarly to other cilia-related proteins, including IFT88, IFT20 participates in the maintenance of energy balance by regulating glucose homeostasis (Figure 3) [91,92,93]. Interestingly, a significant increase in blood glucose levels and impaired glucose tolerance have been observed in tamoxifen-inducible *Ift20* conditional knockout mice where *Ift20* was specifically deleted in mesenchymal stem cells (MSCs) at the postnatal stage [93]. Li and colleagues showed that IFT20 deficiency decreases the expression of genes encoding glycolysis-regulating enzymes, including the glucose transporter Glut1, which is associated with impaired glucose uptake in MSCs. They reported that IFT20 modulates Glut1 expression acting on the TGF-β signaling pathway and regulating pSmad2/3 expression and nuclear localization, thereby uncovering a new role for IFT20 in glucose metabolism.

## 5. Conclusions

While originally identified as a molecule required for ciliogenesis and believed to be specifically expressed by ciliated cells, it is now evident that IFT20 plays a broader physiological role as a central regulator of intracellular trafficking. The emerging scenario is that by regulating intracellular vesicular transport, IFT20 controls crucial processes including PC assembly and ciliary signaling as well as cell activation and migration, autophagy, lysosome biogenesis and function, secretion and glucose metabolism (Figure 3). These findings underscore the importance of IFT20 in cell biology and open new scenarios. Loss of IFT20 has been associated with cystic kidney disease [94] and with achondroplasia [95]. However, based on its cellular functions [34], IFT20 defects may also be associated with different system-related diseases including those affecting the cardiovascular, immune, nervous, reproductive, and respiratory systems. Furthermore, IFT20 may be implicated in the modulation of signaling and cell differentiation of different cell types in addition to those previously described, which include T lymphocytes [37,53,62,65], proepicardium and myocardium cells [72], chondrocytes [96] and osteoblasts [83,89]. Interestingly, it has been recently reported that IFT20 is a critical regulator of bone marrow mesenchymal stem cell (MSC) lineage commitment at an early stage [93]. MSCs are multipotent stromal cells that can differentiate into a variety of cell types, including chondrocytes, osteoblasts, adipocytes, and myoblasts, and assist postnatal organ growth, repair and regeneration [97,98]. Li and colleagues found that loss of IFT20 in MSCs inhibits osteoblast differentiation but promotes adipocyte differentiation. Remarkably, they identify a new role for IFT20 in maintaining energy balance that prevents MSC lineage allocation into adipocytes by regulating glucose metabolism mediated by TGF-β-Smad2/3-Glut1 signaling. Importantly, this recent study identifies IFT20 as a promising drug target for the treatment of bone diseases such as osteoporosis and opens exciting new perspectives on the implication of IFT20 in other metabolic diseases.

IFT20 has also been implicated in cancer. In mouse breast cancer cells, IFT20 inhibits epithelial mesenchymal transition and cell migration [56], while in human osteosarcoma and colorectal cancer cells, it promotes invadopodia formation and tumor invasiveness [66,67]. Interestingly, a positive correlation of IFT20 expression with reduced risk of metastasis and prolonged survival in patients with lung adenocarcinoma associated with specific clinicopathological features has recently been reported [99]. Thus, even though the role of IFT20 in cancer needs to be explored in more depth, these findings suggest that IFT20 could play important roles also during tumorigenesis.

## Figures and Tables

**Figure 1 ijms-23-12147-f001:**
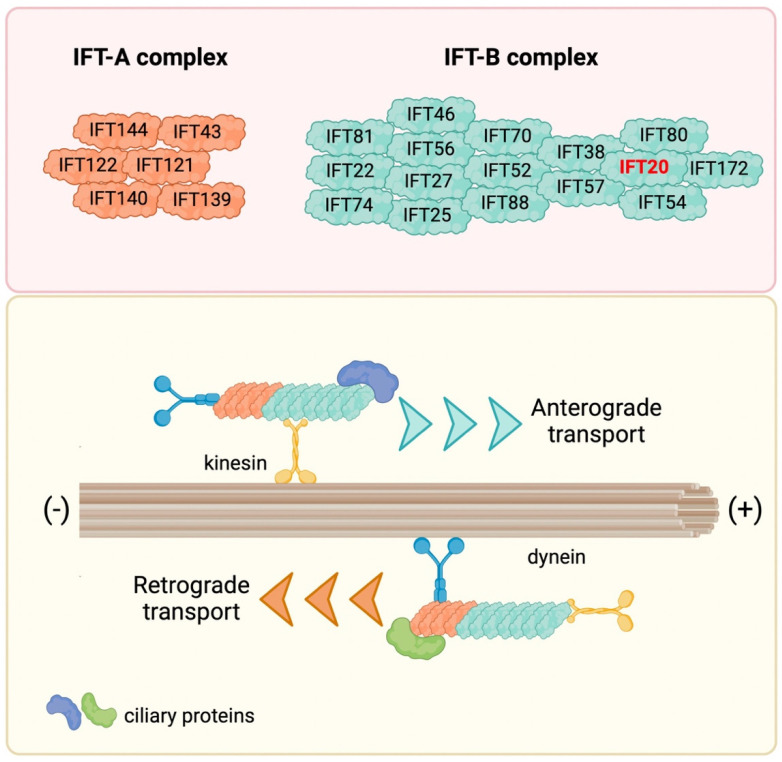
The IFT machinery and its components. IFT proteins are organized in two complexes, namely the IFT-A and IFT-B complex, which carry ciliary proteins along the axonemal microtubules from the cell body to the ciliary tip (anterograde transport) and then back to the cell body (retrograde transport) by interacting with the molecular motors kinesin and dynein, respectively.

**Figure 2 ijms-23-12147-f002:**
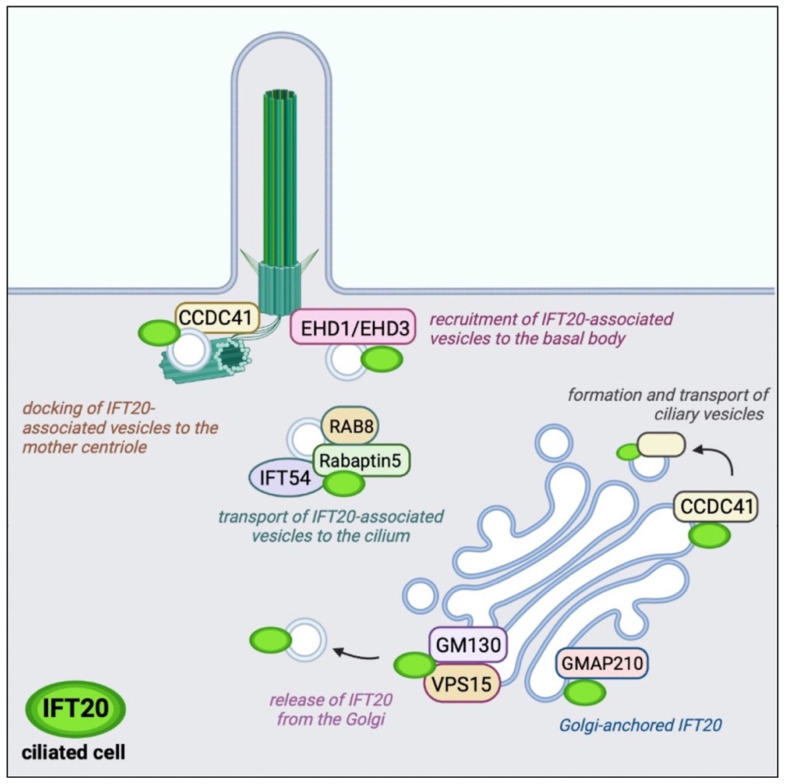
Localization and function of IFT20 interacting proteins in ciliated cells. IFT20 associates with a variety of protein interactors not only at the basal body (EHD1/EHD3) and mother centriole (CCDC41) but also at the Golgi apparatus (GMAP210, CCDC41, GM130/VPS15) and ciliary vesicles (Rab8/Rabaptin5/IFT54) in ciliated cells.

**Figure 3 ijms-23-12147-f003:**
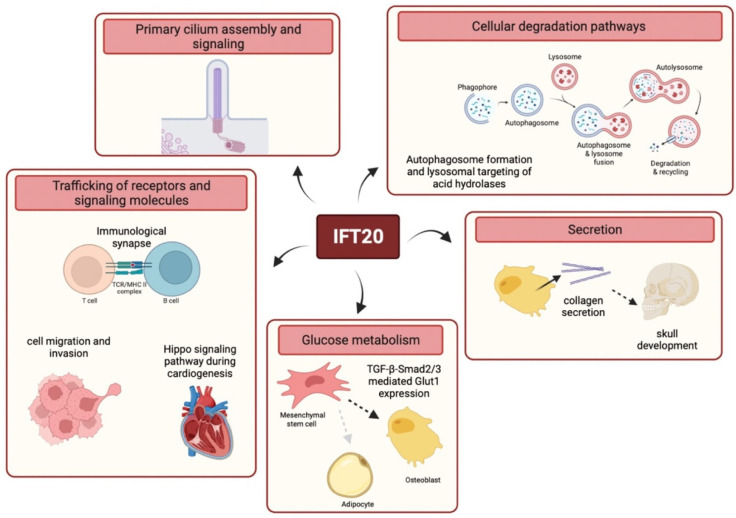
Schematic representation of the main functions of IFT20. IFT20 regulates a number of key processes including PC assembly and signaling, trafficking of receptors and signaling molecules (e.g., at the IS or during cell migration and cardiogenesis) and cellular degradation pathways. Furthermore, IFT20 is implicated in collagen secretion and in Glut1-mediated glucose metabolism during osteoblast differentiation.

## Data Availability

Not applicable.

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
