# Peer review of "IFT20: An Eclectic Regulator of Cellular Processes beyond Intraflagellar Transport"

_ijms, 2022, doi:10.3390/ijms232012147_

Round 1
Reviewer 1 Report
Francesca Finetti and colleagues present a review article on the functions of IFT20, a component of the intraflagellar transport machinery, focusing on its interactions with various other cellular partners regulating intracellular trafficking. In general, the article is comprehensibly written and successfully summarizes most of the acquired knowledge in this field.
Some more articles have very recently appeared in the literature databases and the authors may want to consider including those and discussing the relevant results. A few examples are the following:
doi: 10.1093/biolre/ioab226, doi: 10.1016/j.ydbio.2021.12.004, doi: 10.1002/jcp.30693, doi: 10.1186/s12885-022-09905-6, doi: 10.1002/dvdy.141, doi: 10.1038/s42003-020-0767-x, doi: 10.1016/j.bbamcr.2019.118641, doi: 10.1016/j.ydbio.2022.03.001, doi:10.17912/micropub.biology.000396, doi: 10.1016/j.redox.2021.101969
Although this manuscript mainly focuses on the IFT20 protein in humans, the article would in my opinion benefit from a short paragraph referring to information of homologous proteins in other model systems. Including information on the evolutionary conservation of this protein would also be welcome.
Authors may also want to consider including for the introduction a sketch on the IFT machinery and its components.
Reviewer 2 Report
Very well written and illustrated review on the roles of IFT20 in ciliary trafficking but also in numerous non-ciliary functions (receptor trafficking in non-ciliated cells, secretion of collagen and bone formation, autophagy, cell polarization and differentiation, etc.) in ciliated and non-ciliated cells. A huge synthesis work and an interesting discussion on the potential implication of IFT20 in various diseases based on its cellular roles in different cell types.

Reviewer 3 Report
In their manuscript “IFT20: an eclectic regulator of cellular processes beyond intra-flagellar transport” Finetti et al. provide a detailed and up-to-date review of IFT-20, beyond its function as a component of IFT-B complex.
The manuscript is clear, well-written and well-illustrated. It will be helpful for researchers wanting information on this protein.
Author Response
We thank the Reviewer for his/her positive comments.